# A Novel Nutritional Index Serves as A Useful Prognostic Indicator in Cardiac Critical Patients Requiring Mechanical Circulatory Support

**DOI:** 10.3390/nu11061420

**Published:** 2019-06-24

**Authors:** Asuka Minami-Takano, Hiroshi Iwata, Katsutoshi Miyosawa, Kyoko Kubota, Atsushi Kimura, Shota Osawa, Minako Shitara, Shinya Okazaki, Satoru Suwa, Katsumi Miyauchi, Masataka Sumiyoshi, Atsushi Amano, Hiroyuki Daida

**Affiliations:** 1Department of Cardiovascular Medicine, Graduate School of Medicine, Juntendo University, 2-1-1 Hongo Bunkyo-ku, Tokyo 113-8421, Japan; aminami@juntendo.ac.jp (A.M.-T.); k-miyosawa@juntendo.ac.jp (K.M.); shinya@juntendo.ac.jp (S.O.); ktmmy@juntendo.ac.jp (K.M.); daida@juntendo.ac.jp (H.D.); 2Department of Clinical Engineering, Juntendo University Hospital, 3-1-3 Hongo Bunkyo-ku, Tokyo 113-8431, Japan; s-ohsawa@juntendo.ac.jp (S.O.); me-mina@juntendo.ac.jp (M.S.); 3Department of Clinical Engineering, Juntendo University Shizuoka Hospital, 1129 Nagaoka Izunokuni-city, Shizuoka 410-2295, Japan; kyoko1168@yahoo.co.jp; 4Department of Clinical Engineering, Juntendo University Nerima Hospital, 3-1-10 Takanodai Nerima-ku, Tokyo 177-8521, Japan; a.kimura@juntendo-nerima.jp; 5Department of Cardiovascular Medicine, Juntendo University Shizuoka Hospital, 1129 Nagaoka Izunokuni-city, Shizuoka 410-2295, Japan; ssuwa@juntendo.ac.jp; 6Department of Cardiovascular Medicine, Nerima Hospital, 3-1-10 Takanodai Nerima-ku, Tokyo 177-8521, Japan; sumi@juntendo-nerima.jp; 7Department of Cardiovascular Surgery, Graduate School of Medicine, Juntendo University, 2-1-1 Hongo Bunkyo-ku, Tokyo 113-8421, Japan; a-amano@juntendo.ac.jp

**Keywords:** nutritional index, intensive care unit, mechanical circulatory support, prognostic indicator

## Abstract

Background: A poor nutritional status has been gathering intense clinical interest recently as it has been suggested to associate with adverse outcomes in patients in the intensive care unit (ICU). However, there is still no established nutritional index dominantly used in clinical practice. We have previously proposed a novel nutritional index, which can be calculated using serum levels of triglycerides, total cholesterol, and body weight (TCBI). In this study, to expand the application of TCBI for critical patients, we investigated the usefulness of TCBI to predict prognosis in hemodynamically unstable patients with percutaneously implantable mechanical circulatory support (MCS) devices in the ICU. Patients and Methods: This is a retrospective analysis of a multicenter registry consisting of three Juntendo University hospitals in Japan involving patients who received MCS devices, including intra-aortic balloon pumping (IABP) with or without veno-arterial extracorporeal membrane oxygenation (VA-ECMO), between 2012 and 2016 (*n* = 439). The median follow-up period was 298 days. Results: Spearman’s correlation coefficient between TCBI and the geriatric nutritional risk index (GNRI) was 0.44 (*p* < 0.0001), indicating a moderate positive correlation for these two variables. Unadjusted Kaplan–Meier analysis demonstrated reduced risks of all-cause and cardiovascular mortalities in patients with higher tertiles of TCBI. Furthermore, adjusted multivariate Cox proportional hazard analyses revealed that the highest tertile TCBI was an independent predictor for the reduced risk of all-cause mortality (hazard ratio (HR): 0.22, 95% confidence interval: 0.10–0.48, *p* < 0.0001) and cardiovascular mortality (0.20, 0.09–0.45, *p* < 0.0001). Conclusion: A novel and simple to calculate nutritional index, TCBI, can be applicable as a prognostic indicator in hemodynamically unstable patients requiring MCS devices.

## 1. Introduction

A number of studies have demonstrated that undernutrition or malnutrition is present in a substantial number of patients, ranging from 40% to 80%, with critical illness in intensive care units (ICU) [1], and it has been suggested to be associated with an increased mortality rate [2,3] and longer ICU or hospital stay [4]. Despite such a high prevalence of malnutrition or undernutrition in the ICU, there is a paucity of validated and clinically reliable indices of nutritional status in critical settings in the ICU. Moreover, the various nutritional indices have resulted in inconsistency in the prevalence of malnutrition, thereby leading to the underestimation of its clinical significance. Most previous studies regarding nutritional status in the ICU employed nutritional indices consisting of subjective parameters and answers from questionnaires regarding daily food intake and physical activities, such as the nutrition risk screening score 2002 (NRS) [5], subjective global assessment (SGA) [6], 7-point SGA [7], and malnutrition universal screening tool (MUST) [8]. However, in consideration of the difficulties associated with obtaining accurate information via questionnaires due to the deterioration of consciousness in ICU patients and the chaotic and stressful situation for their families, indices obtained from objective parameters may be appropriate in emergency settings. A number of nutritional indices consisting only of objectively measurable parameters, such as the geriatric nutritional risk index (GNRI) [9], controlling nutritional status (CONUT) score [10], and prognostic nutritional index (PNI) [11], have been used in previous studies and a significant association between poor nutritional status indicated by these indexes and poor clinical outcomes in a wide range of cardiovascular diseases has been reported [12,13,14]. Moreover, the nutrition risk in critically ill (NUTRIC) score, by utilizing objective parameters including two severity indices in critical patients, APPACHE [15] and SOFA [16] score, is a nutritional index which is specified for identifying patients who merit aggressive nutritional intervention in the ICU [17]. However, to date, no study has addressed the association between malnutrition or undernutrition and adverse outcomes in critical patients using objective nutritional indices. Moreover, although these objective nutritional indices are relatively simple compared to conventional subjective indices, they are still too complicated for most clinicians, especially in emergency settings, and currently no objective nutritional index is routinely used in the ICU.

Recently, we have demonstrated the clinical usefulness of a novel nutritional/intrinsic metabolism index, which can be calculated by simple multiplication of the serum values of triglycerides, total cholesterol, and body weight divided by 1000 (TCBI) in patients who underwent percutaneous coronary intervention (PCI). In that population, TCBI was positively correlated with GNRI and lower TCBI, indicating that poor nutritional status predicted an increased risk of cardiovascular mortality [18]. In the present study, to expand the clinical application of this novel and easy to calculate nutritional index for the most critical populations, we evaluated TCBI as a potential nutritional index as well as an indicator for mortality in hemodynamically unstable critical patients who require the placement of percutaneously implantable mechanical circulatory support (MCS) devices, including intra-aortic balloon pumping (IABP) with or without veno-arterial extracorporeal membrane oxygenation (VA-ECMO).

## 2. Patients and Methods

### 2.1. Study Population and Follow-Up Period

This is a retrospective observational cohort analysis of 439 consecutive critical patients with cardiovascular disease, who were placed on mechanical circulatory support devices for any reason, including IABP counterpulsation with or without veno-arterial extracorporeal membrane oxygenation (VA-ECMO) due to a hemodynamically critical condition between January 2012 and December 2016 at three Juntendo University hospitals in Tokyo metropolitan and Shizuoka prefecture, Japan. The IABP counterpulsation devices used were a YAMATO/YAMATO Plus 7.5 French (Datascope, Fairfield, NJ, USA) (*n* = 377) or Corart BP sensor balloon 8Fr (Senko Medical Instrument Mfg. Co., Ltd., Tokyo, Japan) (*n* = 62), while a VA-ECMO device (Terumo emergency bypass system, Terumo Inc., Tokyo, Japan) was placed in 71 patients. No patient received a VA-ECMO without IABP counterpulsation. All mechanical circulatory support device placements were performed by interventional cardiologists or cardiac surgeons under X-ray imaging. Baseline demographic data were obtained before the placement of an MCS, the day of or one day before the induction. Patients were excluded from the analysis regarding TCBI (*n* = 82), when any data component need for TCBI calculation was not available. Similarly, patients without data to calculate GNRI were excluded from the analysis regarding GNRI (*n* = 47) (Appendix A). To compare the nutritional status among different populations with respect to the severity of cardiovascular disease, data regarding nutrition in patients with stable coronary artery disease who underwent elective PCI was extracted from the registry database at Juntendo University Hospital (*n* = 2021). This study was approved by the Institutional Review Board of Juntendo University School of Medicine and is registered in the University Hospital Medical Information Network Clinical Trials Registry (UMIN-CTR) (ID: UMIN000007555). Written informed consent was obtained from all participants, or from their representatives when there was difficulty communicating with the participant. The median and range of the follow-up period after IABP use were 298 days and 0–1869 days, respectively.

### 2.2. Endpoints

The endpoints evaluated were two types of mortality in the follow-up period, all-cause mortality and cardiovascular mortality. Cardiovascular mortality was defined as death due to myocardial infarction, critical arrhythmia, decompensated heart failure, valvular heart disease, an aortic disease, peripheral disease, or sudden cardiac death, in which a non-cardiovascular cause could be excluded. During the follow-up period, all-cause death and cardiovascular death occurred in 112 (25.5%) and 105 (23.9%) out of 439 patients, respectively (and 93.8% was a cardiovascular cause among all-cause deaths). In 105 patients with cardiovascular mortality, 95 (90.5%) died within 100 days after MCS induction.

### 2.3. Nutritional Indices, GNRI, and TCBI

As a control nutritional index for comparison with TCBI, GNRI was calculated in accordance with the following formula as previously reported [9].
GNRI = 14.89 × serum Alb (g/dL) + 41.7 × (measured body weight (kg)/ideal body weight (kg))

Ideal body weight for women was calculated using the Lorentz formula.
Ideal body weight = (height (cm) − 100) − (height (cm) − 150)/4 for men and(height (cm) − 100) − (height (cm) − 150)/2

TCBI was developed as previously described [18]. Briefly, it was defined in accordance with the following principles: (1) objectively measurable; (2) consisted of parameters measured in the majority of patients with cardiovascular disease; and (3) easy to calculate in actual clinical settings, even in emergent cardiovascular situations. Therefore, body weight (BW), instead of body mass index (BMI), was employed as a part of the TCBI as no superiority was addressed by using BMI compared to BW and BMI needs an additional calculation step.
TCBI = Serum level of triglycerides (TG) (mg/dL) × total cholesterol (TC) (mg/dL) × BW (kg)/1000

Participants were divided by the tertiles of GNRI and TCBI. The cut-off levels of GNRI were 89.3 and 98.6 and those of TCBI were 590.0 and 1205.4, respectively.

### 2.4. Statistical Analysis

Continuous valuables are presented as the mean ± standard deviation or median with interquartile range in accordance with the results of the Shapiro–Wilk normality test and were compared using the non-parametric Mann–Whitney test. Comparisons in multiple groups of more than two were performed using 2-way ANOVA followed by multiple post-hoc comparisons by the Bonferroni method. Categorical data are shown as numbers and percentages and were compared using the Fisher exact test. Kaplan–Meier curves for evaluation of the time to the two types of mortality were drawn and followed by the log-rank test for comparison. Unadjusted univariate Cox proportional hazard analyses for all-cause mortality and cardiovascular mortality and multivariate Cox proportional hazard analysis using a model with variables identified in univariate unadjusted analysis and background demographics comparison among TCBI tertiles were performed to identify factors associated with the incidence of mortality. In addition to age and gender, the inclusion criteria of covariates multivariate analysis were a factor, (1) which was associated with all-cause and cardiovascular mortality in univariate Cox proportional hazard analysis with a *p*-value < 0.1, and (2) those who were significantly different in comparisons of background demographics, while BMI, serum albumin, and dyslipidemia were excluded since these parameters were directly associated with TCBI as well as GNRI. Receiver operating characteristic curves were described and areas under the curve (C-statistic) were measured for all-cause and cardiovascular mortalities. All probability values (*p*-values) were two-tailed and considered to be significant if less than 0.05.

## 3. Results

### 3.1. Baseline Patient Demographics in Tertiles of TCBI

Baseline patient demographics before the induction of MCS in accordance with the tertiles of TCBI are listed in Table 1. Patients with lower tertiles were older, more likely to be female, and more likely to have lower blood pressure and chronic kidney disease (CKD) with lower eGFR but less complicated by diabetes and dyslipidemia. Serum albumin and GNRI were lower in patients with lower TCBI. The most frequent cause of MCS induction was acute coronary syndrome (ACS), and the second was cardiogenic shock without significant difference among the groups. The rate of concomitant implantation of V-A ECMO in addition to IABP was 16.1% (*n* = 70), even though there was no difference among TCBI tertiles. The duration of MCS support was similar among the groups. Among lipid parameters, not only serum total cholesterol and triglycerides levels, but also high-density lipoprotein-cholesterol (HDL-C) and low-density lipoprotein-cholesterol (LDL-C) levels were lower in those with lower TCBI tertiles.

### 3.2. Linear Positive Correlation of TCBI with GNRI

The median and interquartile range (IQR) of TCBI in patients with MCS were 835.13 and 481.38–1617.00, respectively, while those in patients with stable angina who underwent elective PCI were higher at 1309.1 and 857.7–2060.7, respectively (*p* < 0.0001). We examined the correlation of TCBI with GNRI in patients with MCS. A correlation analysis followed by Spearman’s nonparametric test indicated the positive correlation and the linear correlation between two variables were moderate in the total participant population (Spearman’s correlation coefficient (*r*): 0.44, *p* < 0.0001). In subpopulations divided by patients with and without ACS, and the gender, TCBI and GNRI were constantly correlated, including in patients without ACS (*r* = 0.25, *p* = 0.02), with ACS (*r* = 0.5, *p* < 0.0001), female (*r* = 0.31, *p* = 0.005) and male (*r* = 0.46, *p* < 0.0001) patients (Figure 1).

### 3.3. Nutritional Status Was Poorer in the Patient Population with More Severe Cardiovascular Diseases

The values of GNRI and TCBI were calculated to compare the nutritional status among different populations regarding the severity of cardiovascular disease, in patients with stable CAD and with those with MCS (Appendix A). Both GNRI and TCBI were significantly lower in the population with MCS than in that with stable CAD (median TCBI 835.1 (IQR 481.4–1620.0) vs. 1329.2 (IQR 886.6–2096.0) (*p* = 0.0003) and mean GNRI 94.9 (±SD 11.7) vs. 100.1 (±SD 7.7) (*p* < 0.0001), respectively). Both GNRI and TCBI were significantly lower in the population with MCS than that of stable CAD. Moreover, in the MCS group, the nutritional indices were significantly lower in patients without ACS (ACS-), such as those with cardiogenic shock, heart failure, and myocarditis, than in patients implanted for ACS (ACS+). These results indicated that more severe patients likely had a lower nutritional status. Moreover, the trends of GNRI and TCBI among the groups were almost identical, indicating these two nutritional indices were similar.

### 3.4. Higher TCBI Was Associated with Lower Incidences of All-Cause and Cardiovascular Mortalities in Unadjusted Kaplan–Meier Analyses

To test the prognostic implication of TCBI, unadjusted Kaplan–Meier analysis was performed. As demonstrated in Figure 2, unadjusted Kaplan–Meier analysis demonstrated a lower cumulative incidence of all-cause and cardiovascular mortality rates in patients with higher TCBI. Log-rank comparisons of each curve for all-cause mortality were significantly different in all comparisons between the groups of TCBI (TCBI- T1 vs. T2: *p* = 0.049, T2 vs. T3: *p* = 0.018, and T1 vs. T3: *p* < 0.0001, respectively), which were similar to those of the analysis stratified by GNRI tertile (Appendix A). Moreover, the cumulative incidence of cardiovascular mortality in TCBI-T3 (highest TCBI group) was significantly lower compared to those in TCBI-T1 and -2 (TCBI-T3 vs. TCBI-T1: *p* < 0.0001, TCBI-T3 vs. T2: *p* = 0.001) while not significantly different between TCBI-2 and -3.

### 3.5. Lower Risk of All-Cause and Cardiovascular Mortality in Patients with Higher TCBI

Unadjusted univariate Cox proportional hazard analyses showed higher TCBI and GNRI were both associated with reduced risk for all-cause mortality and cardiovascular mortality as nominal (TCBI-T3: highest tertile of TCBI) and continuous variables (1 standard deviation (SD) higher TCBI) (Appendix A). Furthermore, multivariate Cox proportional hazard analysis adjusted by variables identified by univariate Cox proportional hazard analysis and background demographics, including age, gender, systolic blood pressure, concomitant use of VA-ECMO, diabetes, chronic kidney disease, acute coronary syndrome, cardiogenic shock, and congestive heart failure, showed an independent association of the higher TCBI with the reduced risk of all-cause and cardiovascular mortalities as a nominal variable and a continuous variable (Figure 3a). These findings were similar to those in the analyses of GNRI (Figure 3b).

### 3.6. Comparable Prognostic Implication of TCBI for All-Cause and Cardiovascular Mortalities with GNRI

We measured the c-statistics, areas under the receiver operating characteristic (ROC) curves of TCBI, GNRI, and BMI for all-cause mortality and found a comparable prognostic implication for TCBI compared to GNRI, while no significant difference in BMI with the reference line was observed (C-statistic: 0.5) (Figure 4a). Similar findings were obtained in the same analysis for cardiovascular mortality (Figure 4b).

### 3.7. Higher TCBI Was Associated with Lower All-Cause Mortality after the Adjustment with Markers of Organ Function

Multivariate Cox proportional hazard analysis adjusted by age and markers of cardiac (BNP), renal (eGFR) and hepatic (total bilirubin) function showed the association of one standard deviation (SD) higher TCBI with reduced risk of all-cause mortality (Appendix A).

## 4. Discussion

This study demonstrated that a novel nutritional and intrinsic metabolism index, TCBI, which can be simply calculated by multiplying TG, TC, and BW, was useful not only as a nutritional index, but also as a prognostic indicator in the most critical populations, such as ICU patients who have a hemodynamically unstable cardiovascular disease requiring percutaneous MCS implantation. TCBI was positively correlated with the conventional but not widely used nutritional index GNRI. Unadjusted Kaplan–Meier analysis, and unadjusted and adjusted Cox proportional hazard analyses constantly demonstrated an association between a higher TCBI and reduced risks of all-cause and cardiovascular mortalities in this critical population. The prognostic implication of TCBI indicated by the C-statistic was comparable with that of GNRI.

The significance of evaluating nutritional status has been gathering clinical attention in a wide range of diseased populations for its usefulness as a prognostic indicator. In the field of cardiovascular diseases, the association between malnutrition or poor nutritional status and adverse outcomes, in other words, an elevated risk of adverse outcomes in undernourished populations, has been reported in patients with coronary artery disease [13,19], heart failure [14,20], peripheral artery disease [21] patients who underwent coronary artery bypass graft surgery [22,23], valvular surgery [24], or trans arterial valve implantation (TAVI) [25]. In patients with critical situations, the adverse prognostic impact of the malnutrition on short-term in-hospital mortality has been evaluated in patients hospitalized in intensive care units (ICU) [1,2]. Although a number of nutritional indices have been proposed in various clinical settings, there is no established nutritional index which is currently routinely used in clinical settings. Nutritional indices in previous studies assessing nutritional status in ICU settings include the nutrition risk screening score 2002 (NRS) [5], subjective global assessment (SGA) [6], 7-point SGA [7], and malnutrition universal screening tool (MUST) [8]. These indices mainly consist of subjective parameters such as answers to questionnaires regarding food intake and physical activity and were obtained by a dietitian from patients and/or their main caregivers within 48 h after admission. However, the collection of precise data regarding nutritional status using questionnaires may be difficult in cases of critically ill patients, especially in emergent situations, due to the variable levels of consciousness in patients and the chaotic and stressful situations of the families or caregivers. On the contrary, a poor nutritional status assessed by nutritional indices with objective parameters, such as GNRI and CONUT, has been recognized as a clinically significant useful indicator of outcomes in various cardiovascular diseases. In addition, the NUTRIC score specified for critical patients assesses the severity of patients for risk stratification to assess the need for aggressive nutritional intervention. However, the prognostic impact of the nutritional status of patients in the most critical and emergent conditions, such as those in the ICU who need MCS devices, has been rarely evaluated. We assessed the nutritional status at the induction of MCS in this study by using objective nutritional indices and found that lower levels of nutritional indices, indicating a poor nutritional status or malnourishment, were associated with increased risk of all-cause mortality and cardiovascular mortality. These findings indicate that objective nutritional indices were useful even in the most critical situations in the ICU for the prediction of their outcomes. Moreover, the prognostic implication of our novel nutritional index, TCBI, was comparable with that of GNRI in this study population, suggesting that it can be applied to critical patients.

As previously described, TCBI was originally developed as a nutritional assessment tool for atherosclerotic cardiovascular disease, in which excessive TG, TC, and body weight (or BMI) are known risk factors for the development of atherosclerosis [18]. The association of low body weight/body mass index with adverse outcomes is known as the “obesity paradox” in various cardiovascular disease populations, which was potentially mediated by poor nutrition and frailty [26,27]. However, the relationship between reduced TG and TC and poor nutritional status was not fully elucidated, although CONUT includes TC in addition to serum albumin and total lymphocyte count and lower TC was associated with a higher CONUT, indicating malnutrition [10]. It should be noted that the intrinsic levels of TG and TC used in TCBI reflect not only the intake, but also the metabolism of fat, cholesterol, carbohydrates, and sugar.

The most significant advantages of TCBI compared to GNRI and CONUT are its simplicity and the time and effort saved when calculating parameters involved in caring for and treating ICU patients. The calculation of TCBI simply requires multiplying 3 cardiology-friendly variables [28], while anyone who calculates GNRI needs to know the constant values 14.89 and 41.7, and how to calculate ideal body weight, which is different among males and females [9]. For CONUT, they need to know thresholds and scores for serum albumin, total lymphocyte counts, and total cholesterol as it is a score scale [10]. Such advantages may be particularly beneficial among medical professionals in the ICU. In this study population, who are more critical compared to those in the previous study, the absolute values of the nutritional indices TCBI and GNRI were significantly lower. These findings may indicate that the baseline nutritional status was lower in the more critical and hemodynamically unstable population requiring MCS devices, compared to those with stable CAD.

This study has several limitations. First, since this multicenter study was a relatively small-scale, retrospective cohort, a larger sample size in a prospective registry in multiple regions and countries may be needed to generalize the findings. Second, we showed the prognostic implication of TCBI in patients with MCS in the present study, however, such patients are generally heterogenous in terms of cardiovascular diseases, such as acute coronary syndrome, heart failure, cardiogenic shock, and myocarditis. Therefore, since the prognostic impact of TCBI may vary among diseases, it may need to be evaluated separately. Third, we did not assess the effects of medications on the prognostic implication of TCBI in this study like in a previous study that showed the usefulness of TCBI in patients taking statins. However, as lipid-lowering medications such as statins, ezetimibe, and PCSK9 inhibitors may have an effect on TCBI values, larger scale studies in critical settings with patients stratified according to whether they receive these lipid-lowering medications or not may be needed. Moreover, the severity indexes in critical patients, APPACHE, SOFA, and NUTRIC scores, were not available in this study. Future studies may be needed to compare the prognostic utility of TCBI with these severity indexes. Fourth, since TCBI was evaluated at the initiation of MCS devices in this study, the prognostic effect of temporal changes in TCBI at multiple time points may be able to address the clinical efficacy of nutritional intervention, which is under intense debate in ICU patients [2]. Finally, the participants in this study underwent implantation of an IABP and/or V-A ECMO, as these two devices were only available in Japan during the 2012–2016 study period. However, temporary ventricular assist devices other than these two, such as Tandemheart and Impella, have been recently implanted in the majority of such patients [29,30] in conjunction with substantial progress in the field of percutaneously implantable MCS devices for hemodynamically unstable patients. Therefore, the prognostic impact of TCBI may also need to be assessed in patients with these devices.

## 5. Conclusions

In conclusion, a novel and simple method to calculate the nutritional index TCBI, obtained by multiplying TG, TC, and BW, was associated with the most often used objective nutritional index, GNRI, in patients with critical and hemodynamically unstable cardiovascular diseases. Moreover, a higher TCBI value was significantly associated with better outcomes and lower all-cause and cardiovascular mortality rates in such a population. These findings indicate that TCBI is applicable as a nutritional and prognostic indicator for patients with critical and hemodynamically unstable cardiovascular diseases requiring MCS devices.

## Figures and Tables

**Figure 1 nutrients-11-01420-f001:**
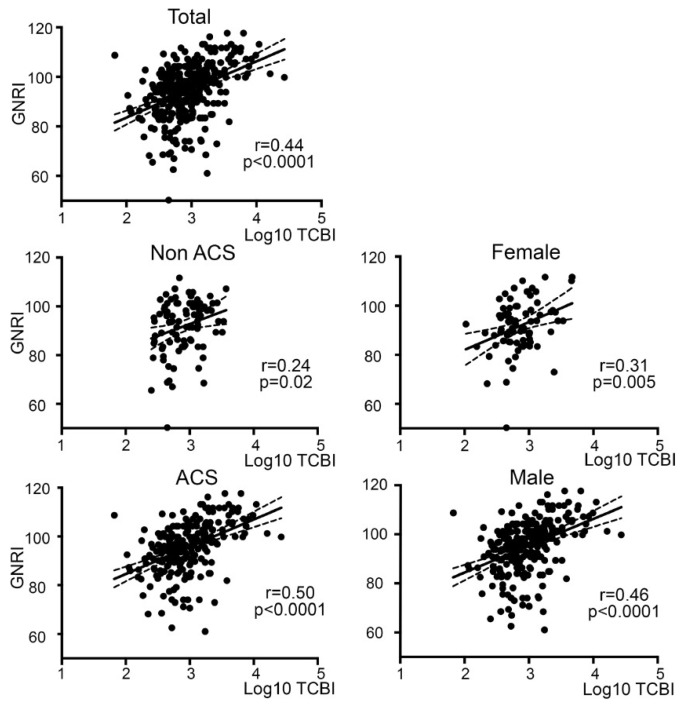
Correlation between TCBI and GNRI in total and subpopulations. Non-parametric Spearman’s correlation coefficients indicated significant, positive and linear correlations between TCBI and GNRI in all populations. r: Correlation coefficient.

**Figure 2 nutrients-11-01420-f002:**
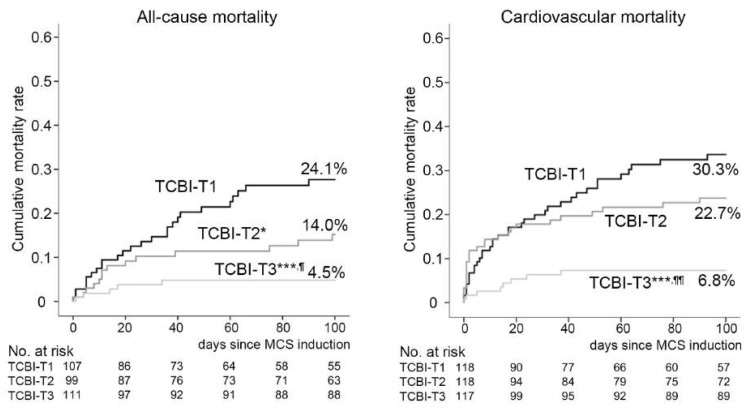
Higher TCBI was associated with reduced risk of all-cause and cardiovascular mortalities. Kaplan–Meier analysis and log-rank comparisons to assess cumulative incidence rates of both all-cause (left panel) and cardiovascular (right panel) mortality in accordance with tertile of TCBI. * <0.05, ** <0.01, *** <0.001 in log-rank comparison vs. TCBI-T1 and ^¶^ <0.05, ^¶¶^ <0.01, ^¶¶¶^ <0.001 vs. TCBI-2.

**Figure 3 nutrients-11-01420-f003:**
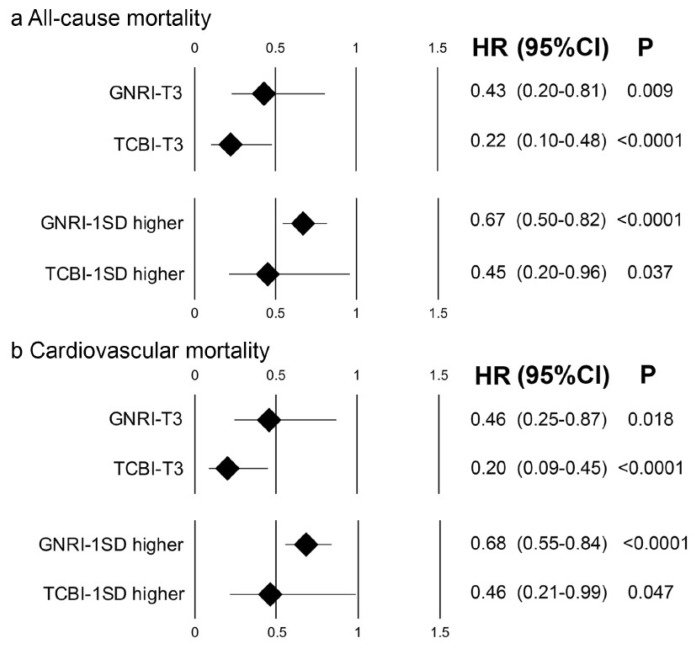
Adjusted hazard ratios for all-cause and cardiovascular mortalities of higher TCBI. Forest plots of higher GNRI and TCBI as a nominal variable (highest tertile, T3, upper panel) and continuous variable (1 standard deviation (SD) higher, lower panel) for all-cause **(a)** and cardiovascular **(b)** mortality. CI: confidence interval, P: *p*-value.

**Figure 4 nutrients-11-01420-f004:**
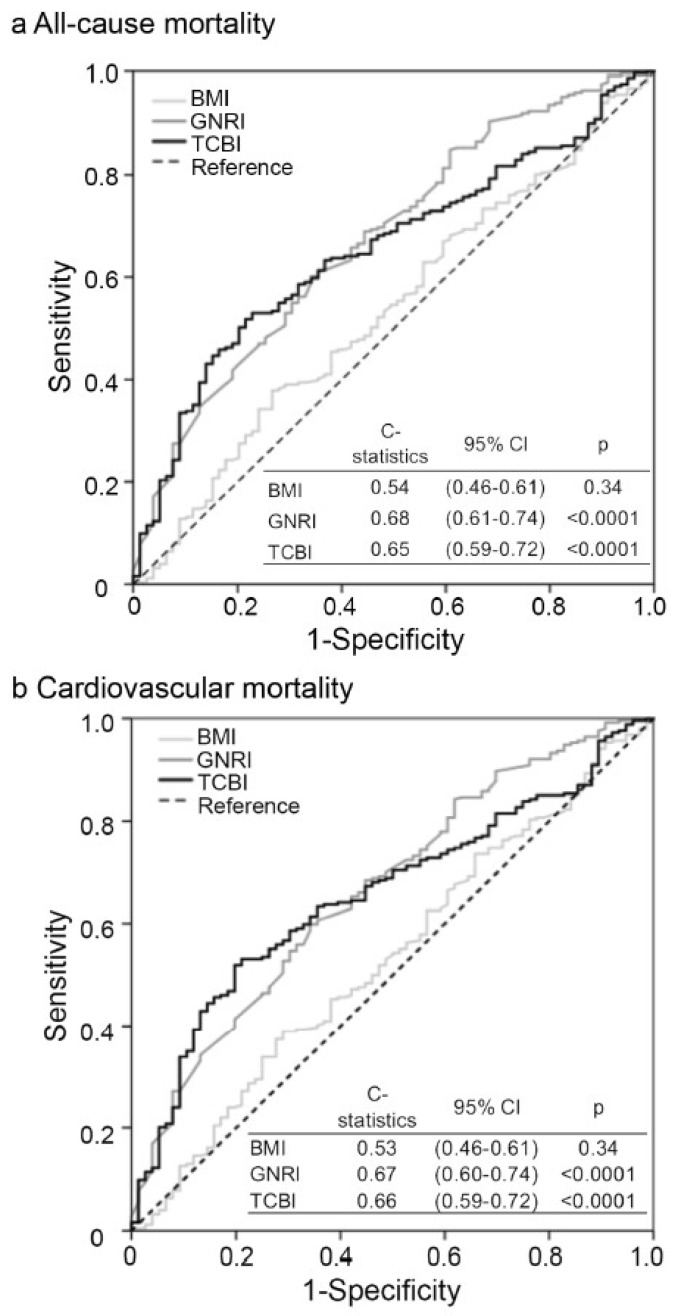
Prognostic implication of BMI, GNRI, and TCBI for all-cause and cardiovascular mortalities. Receiver operating characteristic (ROC) curves of BMI, GNRI, and TCBI with reference line for all-cause **(a)** and cardiovascular **(b)** mortality.

**Table 1 nutrients-11-01420-t001:** Patient demographics according to tertile of triglycerides, total cholesterol, and body weight index (TCBI).

	TCBI T1 (*N* = 119)	TCBI T2 (*N* = 119)	TCBI T3 (*N* = 119)	*p*-value
Age	74.9 ± 9.5	72.2 ± 10.0	64.7 ± 12.0	<0.001
Gender, male	85, 71.4%	88, 74.0%	103, 86.6%	0.010
Body mass index, kg/m^2^	21.6 ± 3.2	22.7 ± 3.5	25.7 ± 4.2	<0.001
Systolic blood pressure, mm Hg	96.7 ± 30.8	102.3 ± 32.1	110.7 ± 29.4	0.003
Heart Rate, beats per minute	86.5 ± 26.4	88.7 ± 23.6	88.0 ± 22.6	0.780
Dyslipidemia	42, 35.3%	47, 39.5%	99, 83.2%	<0.001
Chronic kidney disease (Grade 3–5)	71, 60.0%	74, 62.2%	46, 38.7%	<0.001
eGFR, mL/min/1.732	49.5 ± 29.0	53 ± 28.4	66.8 ± 24.8	<0.001
Diabetes	30, 25.9%	28, 23.9%	50, 42.0%	0.005
CRP, mg/dL	0.50 (0.30–2.40)	0.40 (0.30–1.65)	0.30 (0.30–0.60)	0.127
Serum Albumin, g/dL	3.4 ± 0.7	3.6 ± 0.5	3.9 ± 0.7	<0.001
GNRI	89.6 ± 10.6	94 ± 8.4	99.6 ± 10.4	<0.001
Cause of MCS indication				
Acute coronary syndrome	88, 74.0%	87, 73.1%	96, 80.1%	0.347
Cardiogenic shock	15, 12.6%	7, 5.9%	17, 14.3%	0.086
IABP use	119, 100%	119, 100%	119, 100%	1
V-A ECMO use	20, 17.0%	18, 15.3%	17, 14.3%	0.839
Lipid parameters				
HDL-C, mg/dL	46.6 ± 14.7	47.9 ± 15.5	43.2 ± 12.2	0.035
LDL-C, mg/dL	84.3 ± 30.9	111.0 ± 34.8	132.4 ± 43.4	<0.001
Total cholesterol, mg/dL	142.5 ± 40.7	178.8 ± 36.3	209.8 ± 52.4	<0.001
Triglycerides, mg/dL	48 (38–61)	80 (68–100)	151 (119–209)	<0.001

eGFR: Estimated glomerular filtration rate; CRP: C-reactive protein; GNRI: Geriatric nutritional risk index; IABP: Intraaortic balloon pumping; V-A ECMO: Veno-arterial extracorporeal membrane oxygenation; MCS: Mechanical circulatory support.

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
