# Peer review of "A Novel Nutritional Index Serves as A Useful Prognostic Indicator in Cardiac Critical Patients Requiring Mechanical Circulatory Support"

_nutrients, 2019, doi:10.3390/nu11061420_

Round 1

Reviewer 1 Report

The authors have performed an observational study to evaluate the prognostic utility of the nutritional index, TCBI, in cardiac critical patients. The study is well constructed and appropriate statistical analyses have been carried out. Some concerns remain and should be addressed prior to publication. 

(1) Methods section indicates that a total of 439 patients were recruited for the study. However, Table 1 shows 119 patients present in each tertile making up a total of 357 patients. The reason for this discrepancy is unclear. Please clarify.

(2) GNRI accounts for differences between genders (ideal body weight) whereas TCBI is gender-agnostic. It would be interesting to see how good the correlation is split across gender.

Figure 1 shows correlation between TCBI and GNRI in total patients with and without ACS. In addition to total patients, correlation between the indices should be investigated based on gender (i.e. TCBI vs GNRI correlation analysis for male and female patients).

From Table 1, it appears that close to 70% of patients were male. Is there sufficient statistical power to perform the analysis split by gender?

(3) I suggest combining supplementary figures 3 and 4 with Figures 3 and 4 respectively. An in-place comparison between all-cause and cardiovascular mortality certainly adds value.

(4) Some typographical errors are present in the manuscript and should be corrected.

Author Response

We thank the Editor for the opportunity to revise the manuscript and provide a response to both reviewers while focusing on clarifying the clinical usefulness of TCBI.

Changes to the manuscript have been indicated by responses in red font. Deleted text is shown in strikethrough font. A clean copy of the revision has been submitted. In the responses to the reviewer comments, the reviewer comments are indicated as bold.

Comments and Suggestions for Authors

The authors have performed an observational study to evaluate the prognostic utility of the nutritional index, TCBI, in cardiac critical patients. The study is well constructed and appropriate statistical analyses have been carried out. Some concerns remain and should be addressed prior to publication. 

All authors thank this reviewer for the positive feedback and constructive comments to help improve this study.

(1) Methods section indicates that a total of 439 patients were recruited for the study. However, Table 1 shows 119 patients present in each tertile making up a total of 357 patients. The reason for this discrepancy is unclear. Please clarify.

We thank the reviewer for the opportunity to address the confusion regarding the participants of the study. This registry database involved 439 patients in total, while data for triglycerides, total cholesterol and body weight were not available in 7, 74 and 8 patients, respectively. Therefore, 82 patients were excluded from the analysis regarding TCBI. Similarly, data of GNRI was missing in 47 patients and they were excluded from the analysis regarding the analysis of GNRI. We have prepared a consort diagram as Supplementary Figure 1, and the Study population and follow-up section was revised.

Supplementary Figure 1: Consort diagram of the study (Please see the attached file)

Revised Study population and follow-up period section (page 7):

Patients were excluded from the analysis regarding TCBI (n=82) since no data components needed for TCBI calculation were available. Similarly, patients without data for calculating GNRI were excluded from the analysis regarding GNRI (n=47) (Supplementary Figure 1).

(2) GNRI accounts for differences between genders (ideal body weight) whereas TCBI is gender-agnostic. It would be interesting to see how good the correlation is split across gender. Figure 1 shows correlation between TCBI and GNRI in total patients with and without ACS. In addition to total patients, correlation between the indices should be investigated based on gender (i.e. TCBI vs GNRI correlation analysis for male and female patients). From Table 1, it appears that close to 70% of patients were male. Is there sufficient statistical power to perform the analysis split by gender?

We appreciate the opportunity to evaluate this interesting and clinically relevant point. We analyzed the correlation between TCBI and GNRI separately in female and male patients. Interestingly, TCBI and GNRI were significantly and positively correlated in both female and male patients (correlation coefficient (r): 0.31 and 0.46, p-value: 0.005 and <0.0001, respectively), while r was higher in male patients. Accordingly, we modified Figure 1 to add these findings.

Revised Figure 1: (please see the attached file) 

Revised Results (page 10):

Linear positive correlation of TCBI with GNRI

The median and interquartile range (IQR) of TCBI in patients with MCS were 835.13 and 481.38-1617.00, respectively, while those in patients with stable angina who underwent elective PCI were higher, 1309.1 and 857.7-2060.7, respectively (p<0.0001). We examined the correlation of TCBI with GNRI in patients with MCS. A correlation analysis followed by Spearman’s nonparametric test indicated the positive and the linear correlation between two variables were moderate in the total participant population (Spearman’s correlation coefficient (r): 0.44, p<0.0001). In subpopulations divided by patients with and without ACS, and gender, TCBI and GNRI were constantly correlated, including in patients without ACS (r=0.25, p=0.02), with ACS (r=0.5, p<0.0001), female patients (r=0.31, p=0.005) and male patients (r=0.46, p<0.0001) (Figure 1).

Revised Figure 1 legend (page 21):

Figure 1: Correlation between TCBI and GNRI in total and subpopulations

Non-parametric Spearman’s correlation coefficients indicated significant, positive and linear correlations between TCBI and GNRI in all populations. r: correlation coefficient.

(3) I suggest combining supplementary figures 3 and 4 with Figures 3 and 4 respectively. An in-place comparison between all-cause and cardiovascular mortality certainly adds value.

The authors thank the reviewer for the constructive suggestion. Supplementary Figure 3 now includes Figure 3 and Supplementary Figure 4 includes Figure 4, respectively. Therefore, the Results and Figure 3 and 4 legends were revised.

Revised Figure 3 and 4 (Please see the attached file)

Revised results (page 11 and page 12):

page11

Furthermore, multivariate Cox proportional hazard analysis adjusted by variables identified by univariate Cox proportional hazard analysis and background demographics, including age, gender, systolic blood pressure, concomitant use of VA-ECMO, diabetes, chronic kidney disease, acute coronary syndrome, cardiogenic shock and congestive heart failure, showed an independent association of the higher TCBI with the reduced risk of all-cause and cardiovascular mortalities as a nominal and a continuous variable (Figure 3a). These findings were similar to those in the analyses of GNRI (Figure 3b).

page 12:

We measured the c-statistics, areas under the receiver operating characteristic (ROC) curves of TCBI, GNRI, and BMI for all-cause mortality and found a comparable prognostic implication of TCBI compared to GNRI, while there was no significant difference in BMI with the reference line (C-statistic: 0.5) (Figure 4a). Similar findings were obtained in the same analysis for cardiovascular mortality (Figure 4b).

Revised Figure 3 legend (page 21):

Figure 3: Adjusted hazard ratios for all-cause and cardiovascular mortalities of higher TCBI 

Forest plots of higher GNRI and TCBI as a nominal variable (highest tertile, T3, upper panel) and continuous variable (1 standard deviation (SD) higher, lower panel) for all-cause (a) and cardiovascular (b) mortality. CI: confidence interval, P: p-value.

Revised Figure 4 legend (page 21):

Figure 4: Prognostic implication of BMI, GNRI, and TCBI for all-cause and cardiovascular mortalities

Receiver operating characteristic (ROC) curves of BMI, GNRI, and TCBI with the reference line for all-cause (a) and cardiovascular (b) mortality.

(4) Some typographical errors are present in the manuscript and should be corrected.

We thank the reviewer for pointing out the typographical errors in the manuscript. We have carefully checked and revised the manuscript to eliminate these errors.

We thank this reviewer for these constructive comments and suggestions which serve to improve the manuscript.

Reviewer 2 Report

I read with interest the study by Minami-Takano et al which examines the prognostic value of the TCBI index in critically ill cardiac patients requiring mechanical circulatory support. The impact of nutrition in this specialized subgroup of critically ill patients is understudied and may have important implications for patient care. Nonetheless, I have a number of questions/concerns regarding their manuscript:

The authors refer to the TCBI as a nutritional index but it seems to be more of a      lipid/obesity index. The WHO definition of malnutrition refers to deficiencies, excesses, or imbalances in      a person’s intake of energy and/or nutrients but not to intrinsic levels      of nutrients per se. These intrinsic levels used in the TCBI may reflect      metabolism as well as intake.

The NUTRIC score is a widely used and recommended (ASPEN Guidelines) measure of nutritional      status in critically ill patients but is not mentioned in the manuscript.      It is based on variables that are readily available/calculated on ICU      admission. The TCBI is compared to the GNRI which is not widely used. Please      explain why NUTRIC was not mentioned in the manuscript or used for comparison      

A large number of biochemical markers and scoring systems have been developed for  prognostication in critically ill patients. If the value of the TCBI is      prognostication then it should be compared to other prognostic scores not nutritional      scores that were not designed for that purpose. Furthermore, the      additional value (if any) for prognostication provided by TCBI after      controlling for well-established predictors (e.g. age, organ dysfunction) is      unknown. This limitation should be mentioned in the manuscript

The manuscript  would benefit by grammatical editing by a native English speaker

From a  clinical standpoint how should the TCBI be used. Why factors can be      modified to reduce risk?

What are the  next steps for TCBI investigation?

Author Response

All authors wish to thank this reviewer for his/her positive critique and constructive comments on our study, which helped to greatly improve this study. Changes to the manuscript have been indicated by responses in red font. Deleted text has been shown in strikethrough font.

Comments and Suggestions for Authors

I read with interest the study by Minami-Takano et al which examines the prognostic value of the TCBI index in critically ill cardiac patients requiring mechanical circulatory support. The impact of nutrition in this specialized subgroup of critically ill patients is understudied and may have important implications for patient care. Nonetheless, I have a number of questions/concerns regarding their manuscript:

The authors refer to the TCBI as a nutritional index but it seems to be more of a lipid/obesity index. The WHO definition of malnutrition refers to deficiencies, excesses, or imbalances in a person’s intake of energy and/or nutrients but not to intrinsic levels of nutrients per se. These intrinsic levels used in the TCBI may reflect metabolism as well as intake.

We appreciate this on-point comment. We agree with the reviewer’s critique that the components in calculating TCBI, especially intrinsic serum levels of triglycerides and total cholesterol, reflect not only the intake, but also the metabolism of fat, cholesterol, carbohydrates and sugar. Moreover, how much both factors influence TCBI levels is undetermined for now. Accordingly, we changed the description of TCBI in the Introduction and Discussion, and discussed this point in the Discussion.

Revised Introduction (page 5):

Recently, we have demonstrated the clinical usefulness of a novel nutritional/metabolism index, which is calculable by simple multiplication of the serum values of triglycerides, total cholesterol and body weight divided by 1000 (TCBI) in patients who underwent percutaneous coronary intervention (PCI).

Revised Discussion (page 13):

This study demonstrated that a novel nutritional and metabolism index, TCBI, which can be simply calculated by multiplying TG, TC and BW, was useful not only as a nutritional index, but also as a prognostic indicator in the most critical populations, such as ICU patients who have a hemodynamically unstable cardiovascular disease requiring percutaneous MCS implantation.

Revised Discussion section (page 14):

It should be noted that the intrinsic levels of TG and TC used in TCBI reflect not only the intake, but also the metabolism of fat, cholesterol, carbohydrates, and sugar.

The NUTRIC score is a widely used and recommended (ASPEN Guidelines) measure of nutritional status in critically ill patients but is not mentioned in the manuscript. It is based on variables that are readily available/calculated on ICU admission. The TCBI is compared to the GNRI which is not widely used. Please explain why NUTRIC was not mentioned in the manuscript or used for comparison.

Thank you for giving us the opportunity to improve this study by providing further discussion regarding nutritional indices in critical patients in the ICU, and we agree that the NUTRIC score is one of the important scores regarding nutrition in ICU patients and needs to be discussed in this study. However, as the APPACHE II and SOFA scores were not designated to be included in this registry database, the NUTRIC score was unfortunately not calculable in this study. Therefore, we have mentioned this in the Introduction and Limitation. However, although we agree with the reviewer that the NUTRIC score is important, we think there is a substantial difference in the significance of NUTRIC and other nutritional scores, including TCBI. For calculating the NUTRIC score, the following parameters are needed; age, number of comorbidities, days of hospital stay before ICU admission, APACHE II score (history of organ failure or immunocompromise, age, temperature, mean arterial pressure, pH, heart rate, respiratory rate, sodium, potassium, creatinine, acute renal failure, hematocrit, white blood cell count, Glasgow coma scale, FiO2), and SOFA score (PaO2, FiO2, mechanical ventilation, platelets, Glasgow coma scale, bilirubin, mean arterial pressure, serum creatinine). The NUTRIC score is for making a decision on nutritional strategy and to identify individuals who benefit from aggressive nutritional intervention by evaluating the severity of patients (Crit Care. 2011;15(6):R268.) rather than their current nutritional status. Accordingly, it does not include any parameter directly associated with the current nutritional status. Conversely, TCBI has been developed as an indicator of the present nutritional (and the metabolic) status. Therefore, we believe that the comparison between TCBI and a conventional nutritional indicator GNRI makes more sense, rather than that with severity indexes. 

Revised Introduction (page 5):

Moreover, the nutrition risk in critically ill (NUTRIC) score by utilizing objective parameters including two severity indices in critical patients, the APPACHE15 and SOFA16 scores, is a nutritional index which is specified for identifying patients who merit aggressive nutritional intervention in the ICU17.

Revised Discussion (page 14):

In addition, the NUTRIC score specified for critical patients, assesses the severity of patients for risk stratification to assess the need for aggressive nutritional intervention.

Revised Limitation (in Discussion) (page 15):

Third, we did not assess the effects of medications on the prognostic implication of TCBI in this study like in a previous study that showed the usefulness of TCBI in patients taking statins. However, as lipid-lowering medications such as statins, ezetimibe and PCSK9 inhibitors may have an effect on TCBI values, larger scale studies in critical settings with patients stratified according to whether they receive these lipid-lowering medications or not may be needed. Moreover, the severity indexes in critical patients, the APPACHE, SOFA and NUTRIC scores, were not available in this study. Future studies may be needed to compare the prognostic utility of TCBI with these severity indexes.

A large number of biochemical markers and scoring systems have been developed for  prognostication in critically ill patients. If the value of the TCBI is prognostication then it should be compared to other prognostic scores not nutritional scores that were not designed for that purpose. Furthermore, the additional value (if any) for prognostication provided by TCBI after controlling for well-established predictors (e.g. age, organ dysfunction) is unknown. This limitation should be mentioned in the manuscript

We thank the reviewer for the opportunity to improve this study by addressing this key issue regarding the clinical usefulness of TCBI. As mentioned above, our registry database did not have APACHE and SOFA severity scores. However, as suggested, to compare the prognostic implication of TCBI with biomarkers which reflect cardiac, renal and hepatic functions, which have been previously reported to be associated with prognosis in critically ill patients, we performed Cox proportional hazard analysis using a model including age, TCBI, plasma brain natriuretic peptide (BNP), and serum creatinine and serum total bilirubin levels. These parameters were evaluated as continuous variables. Consequently, TCBI (1 standard deviation higher), eGFR (10 ml/min/1.73*2 higher) and total bilirubin (1mg/dl higher) were independent predictors for all-cause mortality, while no significant association was found in plasma BNP (500 pg/dl higher) and age (1 year older). These findings indicate that the prognostic implication of TCBI may be at least comparable with biomarkers representing organ function (or dysfunction). These results are now shown in Supplementary Table 2.

Revised Results (page 12):

Higher TCBI was associated with lower all-cause mortality after the adjustment with markers of organ function

Multivariate Cox proportional hazard analysis adjusted by age and markers of cardiac (BNP), renal (eGFR) and hepatic (total bilirubin) function showed the association of one standard deviation (SD) higher TCBI with reduced risk of all-cause mortality (Supplementary Table 2).  

Supplementary Table 2 (page 22):

HR

95% CI

p-value

Age, 1 year older

1.00

0.97-1.03

0.86

TCBI, 1SD higher

0.29

0.10-0.81

0.02

BNP, 100 pg/ml higher

0.96

0.86-1.07

0.44

eGFR, 10 ml/min/1.73m2   higher

0.82

0.73-0.93

0.001

Total bilirubin, 1 mg/dl higher

1.81

1.11-2.95

0.02

The manuscript  would benefit by grammatical editing by a native English speaker

We thank the reviewer for his/her comments regarding English grammar. We have carefully checked and revised the manuscript to eliminate grammatical errors. 

From a clinical standpoint how should the TCBI be used. Why factors can be modified to reduce risk?

What are the  next steps for TCBI investigation?

Thank you for asking very constructive questions regarding practical aspects and future directions of TCBI. Because it is simple to calculate and uses objective parameters measured in the vast majority of patients with cardiovascular disease, TCBI can be applied to patients with various cardiovascular diseases in various stages. Since there was an association between lower TCBI and increased risk of cardiovascular mortalities in patients with coronary artery disease in a previous study (Int J Cardiol. 2018;262:92-98.) and critical patients with mechanical circulatory support device in this study, TCBI may be worth evaluating in other various cardiovascular disease contexts and stages, such as heart failure, aortic disease, and pre- and postsurgical status. Moreover, as mentioned in the Discussion, the prognostic impact of a temporary change in TCBI over multiple time points in the clinical course needs to be also evaluated. Given that an increase in TCBI over time predicts a favorable prognosis, the clinical benefit of nutritional intervention guided by TCBI will be of clinical value.

We thank this reviewer for their constructive comments and suggestions which helped to improve the manuscript.

Round 2

Reviewer 2 Report

The authors have comprehensively addressed my questions and concerns. I have no further suggestions for improvement